# A framework for bilevel optimization that enables stochastic and global variance reduction algorithms

**Mathieu Dagréou**
Inria, CEA
Université Paris-Saclay
Palaiseau, France
`mathieu.dagreou@inria.fr`

**Pierre Ablin**
CNRS
Université Paris-Dauphine, PSL-University
Paris, France
`pierre.ablin@cnrs.fr`

**Samuel Vaiter**
CNRS
Université Côte d'Azur, LJAD
Nice, France
`samuel.vaiter@cnrs.fr`

**Thomas Moreau**
Inria, CEA
Université Paris-Saclay
Palaiseau, France
`thomas.moreau@inria.fr`

## Abstract

Bilevel optimization, the problem of minimizing a *value function* which involves the arg-minimum of another function, appears in many areas of machine learning. In a large scale empirical risk minimization setting where the number of samples is huge, it is crucial to develop stochastic methods, which only use a few samples at a time to progress. However, computing the gradient of the value function involves solving a linear system, which makes it difficult to derive unbiased stochastic estimates. To overcome this problem we introduce a novel framework, in which the solution of the inner problem, the solution of the linear system, and the main variable evolve at the same time. These directions are written as a sum, making it straightforward to derive unbiased estimates. The simplicity of our approach allows us to develop global variance reduction algorithms, where the dynamics of all variables is subject to variance reduction. We demonstrate that SABA, an adaptation of the celebrated SAGA algorithm in our framework, has $O(\frac{1}{T})$ convergence rate, and that it achieves linear convergence under Polyak-Łojasciewicz assumption. This is the first stochastic algorithm for bilevel optimization that verifies either of these properties. Numerical experiments validate the usefulness of our method.

## 1 Introduction

Bilevel optimization is attracting more and more attention in the machine learning community thanks to its wide range of applications. Typical examples are hyperparameters selection [5, 38, 17, 6], data augmentation [11, 42], implicit deep learning [3] or neural architecture search [33]. Bilevel optimization aims at minimizing a function whose value depends on the result of another optimization problem:

$$\min_{x \in \mathbb{R}^d} h(x) = F(z^*(x), x), \quad \text{such that } z^*(x) \in \arg\min_{z \in \mathbb{R}^p} G(z, x) \ , \tag{1}$$

where $F$ and $G$ are two real valued functions defined on $\mathbb{R}^p \times \mathbb{R}^d$. $G$ is called the *inner function*, $F$ is the *outer function* and $h$ is the *value function*. Similarly, $z$ is the *inner variable* and $x$ is the *outer variable*. In most cases, the function $z^*$ can only be approximated by an optimization algorithm, which makes bilevel optimization problems challenging. Under appropriate hypotheses, the function $h$ is differentiable, and the chain rule and implicit function theorem give for any $x \in \mathbb{R}^d$

$$\nabla h(x) = \nabla_2 F(z^*(x), x) + \nabla_{21}^2 G(z^*(x), x) v^*(x) \ , \tag{2}$$

36th Conference on Neural Information Processing Systems (NeurIPS 2022).

where $v^*(x) \in \mathbb{R}^p$ is the solution of a linear system

$$v^*(x) = -\left[\nabla_{11}^2 G(z^*(x), x)\right]^{-1} \nabla_1 F(z^*(x), x) \ . \tag{3}$$

In the light of (2) and (3), it turns out that the derivation of the gradient of $h$ at each iteration is cumbersome because it involves two subproblems: the resolution of the inner problem to find an approximation of $z^*(x)$ and the resolution of a linear system to find an approximation of $v^*(x)$. It makes the practical implementation of first order methods like gradient descent for (1) challenging.

As is the case in many machine learning problems, we suppose in this paper that $F$ and $G$ are empirical means:

$$F(z, x) = \frac{1}{m} \sum_{j=1}^m F_j(z, x), \quad G(z, x) = \frac{1}{n} \sum_{i=1}^n G_i(z, x) \ .$$

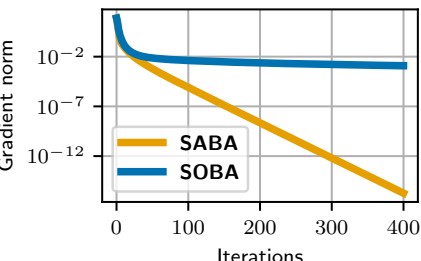

This structure suggests the use of stochastic methods to solve (1). For single-level problems (that is, classical optimization problems where one function should be minimized), using Stochastic Gradient Descent (SGD; [41, 7]) and variants is natural because individual gradients are straightforward unbiased estimators of the gradient. In the bilevel framework, we want to develop algorithms that make progress on problem (1) by using only a few functions $F_j$ and $G_i$ at a time. However, since $\nabla h$ involves the inverse of the Hessian of $G$, building such stochastic

Figure 1: Convergence curves of the two proposed methods on a toy problem. SABA is a stochastic method that achieves fast convergence on the value function.

algorithms is quite challenging, one of the difficulties being that there is no straightforward unbiased estimator of $\nabla h$. Still, in settings where $m$ or $n$ are large, where computing even a single evaluation of $F$ or $G$ is extremely expensive, stochastic methods are the only scalable algorithms.

Variance reduction [27, 13, 43, 15, 12] is a popular technique to obtain fast stochastic algorithms. In a single-level setting, these methods build an approximation of the gradient of the objective function using only stochastic gradients. Contrary to SGD, the variance of the approximation goes to 0 as the algorithm progresses, allowing for faster convergence. For instance, the SAGA method [13] achieves linear convergence if the objective function satisfies a Polyak-Łojasciewicz inequality, and $O(\frac{1}{T})$ convergence rate on smooth non-convex functions [40]. The extension of these methods to bilevel optimization is a natural idea to develop faster algorithms. However, this idea is hard to implement because it is hard to derive unbiased estimators of $\nabla h$, let alone variance reduction ones.

**Contributions.** We introduce a **novel framework for bilevel optimization** in Section 2, where the inner variable, the solution of the linear system (3) and the outer variable evolve jointly. The evolution directions are written as sums of derivatives of $F_j$ and $G_i$, which allows us to derive simple unbiased stochastic estimators. In this framework, we propose SOBA, an extension of SGD (Section 2.1), and SABA (Section 2.2), an extension of the variance reduction algorithm SAGA [13]. In Section 3 we analyse the convergence of our methods. SOBA is shown to achieve $\inf_{t \leq T} \mathbb{E}[\|\nabla h(x^t)\|^2] = O(\log(T)T^{-\frac{1}{2}})$ with decreasing step sizes. We prove that SABA with fixed step sizes achieves $\frac{1}{T} \sum_{t=1}^T \mathbb{E}[\|\nabla h(x^t)\|^2] = O(\frac{1}{T})$. SABA is therefore, to the best of our knowledge, the first **stochastic bilevel algorithm that matches the convergence rate of gradient descent** on $h$. We also prove that SABA achieves **linear convergence** under the assumption that $h$ satisfies a Polyak-Łojasciewicz inequality. To the best of our knowledge, SABA is also the first stochastic bilevel algorithm to feature such a property. Importantly, **these rates match the rates of the single level counterparts of each algorithm in non-convex setting** (SGD for SOBA and SAGA for SABA). Finally, in Section 4, we provide an **extensive benchmark** of many stochastic bilevel methods on hyperparameters selection and data hyper-cleaning, and illustrate the usefulness of our approach.

**Related work.** The bilevel optimization problem has a strong history in the optimization community, taking root in game theory [45]. Gradient-based algorithms to solve (1) can be mainly classified in two different categories depending on how $\nabla h$ is computed, by *automatic* or *implicit differentiation*.

Since the solution of the inner problem $z^*(x)$ is approximated by the output of an iterative algorithm, it is possible to use automatic differentiation [46, 31] to approximate $\nabla h(x)$. It consists in differentiating the different steps of the inner optimization algorithm – see [4] for a review – and has been applied successfully to several bilevel problems arising in machine learning [14, 16]. One

of the main drawbacks of this approach is that it requires to store in memory each iterate of the inner optimization algorithm, although this problem can sometimes be overcome using invertible optimization algorithms [34] or truncated backpropagation [44].

The use of the implicit function theorem to obtain (2) and (3) is known as implicit differentiation [5]. While the cost of computing exactly (2) can be prohibitive for large scale problems, Pedregosa [38] showed that we can still converge to a stationary point of the problem by using approximate solutions of the inner problem and linear system (3), if the approximation error goes to 0 sufficiently quickly. The complexity of approximate implicit differentiation has been studied in [20]. Ramzi et al. [39] propose to reuse the computations done in the forward pass to approximate the solution of the linear system (3) when the inner problem is solved thanks to a quasi-Newton method.

In the last few years, several works have proposed different strategies to solve (1) in a stochastic fashion. A first set of methods relies on *two nested loops*: one inner loop to solve the inner problem with a stochastic method, and one outer loop to update the outer variable with an approximate gradient direction. In [19, 26, 9] the authors use several SGD iterations for the inner problem and then use stochastic Neumann approximations to get an estimate solution of the linear system, which provides them with an approximation of $\nabla h$ used to update $x$. The analysis of this kind of method was refined by Chen et al. [9], allowing to achieve the same convergence rates as those of SGD. The convergence of the hypergradient when using stochastic solvers for the inner problem and the linear system has been studied in [21]. Arbel and Mairal [2] replace the Neumann approximation by SGD steps to estimate (3). Other authors have proposed *single loop* algorithms, alternating steps in the inner and the outer problem. Hong et al. [24] propose to perform Neumann approximations of the inverse Hessian and use a single SGD step for the inner problem. It was refined in [23] and [47] where the optimization procedure uses a momentum acceleration. Other variations around this idea include [25, 28, 10, 22, 30]. We refer to Table 1 in appendix for a detailed comparison of these methods.

**Notation.** The set of integers between 1 and $n$ (included) is denoted $[n]$. For $f : \mathbb{R}^p \times \mathbb{R}^d \to \mathbb{R}$, we denote $\nabla_i f(z, x)$ its gradient w.r.t. the $i^{\text{th}}$ variable. The Hessian of $f$ with respect to the first variable is denoted $\nabla_{11}^2 f(z, x) \in \mathbb{R}^{p \times p}$, and the cross-derivatives matrix is $\nabla_{21}^2 f(z, x) \in \mathbb{R}^{d \times p}$. If $v$ is a vector, $\|v\|$ is its Euclidean norm. If $M$ is a matrix, $\|M\|$ is its spectral norm. A function is said to be $L$-smooth, for $L > 0$, if it is differentiable, and its gradient is $L$-Lipschitz.

## 2   Proposed framework

In this section, we introduce our framework in which the solution of the inner problem, the solution of the linear system (3) and the outer variable all evolve at the same time, following directions that are written as a sum of derivatives of $F_j$ and $G_i$. We define

$$D_z(z, v, x) = \nabla_1 G(z, x) \ , \tag{4}$$

$$D_v(z, v, x) = \nabla_{11}^2 G(z, x)v + \nabla_1 F(z, x) \ , \tag{5}$$

$$D_x(z, v, x) = \nabla_{21}^2 G(z, x)v + \nabla_2 F(z, x) \ . \tag{6}$$

These directions are motivated by the fact that we have $\nabla h(x) = D_x(z^*(x), v^*(x), x)$, with $z^*(x)$ the minimizer of $G(\cdot, x)$ and $v^*(x)$ the solution of $\nabla_{11}^2 G(z^*(x), x)v = -\nabla_1 F(z^*(x), x)$. When $x$ is

---

**Algorithm 1** General framework

**Input:** initializations $z_0 \in \mathbb{R}^p$, $x_0 \in \mathbb{R}^d$, $v_0 \in \mathbb{R}^p$, number of iterations $T$, step size sequences $(\rho^t)_{t<T}$ and $(\gamma^t)_{t<T}$.
**for** $t = 0, \dots, T-1$ **do**
    Update $z$: $z^{t+1} = z^t - \rho^t D_z^t$ ,
    Update $v$: $v^{t+1} = v^t - \rho^t D_v^t$ ,
    Update $x$: $x^{t+1} = x^t - \gamma^t D_x^t$ ,
    where $D_z^t, D_v^t$ and $D_x^t$ are unbiased estimators of $D_z(z^t, v^t, x^t), D_v(z^t, v^t, v^t)$ and $D_x(z^t, v^t, x^t)$.
**end for**

---

fixed, we approximate $z^*$ by doing a gradient descent on $G$, following the direction $-D_z(z, v, x)$. Finally, when $z$ and $x$ are fixed, we find $v^*$ by following the direction $-D_v(z, v, x)$, which corresponds to a gradient descent on $v \mapsto \frac{1}{2} \langle \nabla_{11}^2 G(z, x)v, v \rangle + \langle \nabla_1 F(z, x), v \rangle$. The rest of the paper is devoted to the study of the global dynamics where the three variables $z, v$ and $x$ evolve at the same time, following stochastic approximations of $D_z, D_v$ and $D_x$. The next proposition motivates the choice of these directions.

**Proposition 2.1.** *Assume that for all $x \in \mathbb{R}^d$, $G(\cdot, x)$ is strongly convex. If $(z, v, x)$ is a zero of $(D_z, D_v, D_x)$, then $z = z^*(x)$, $v = v^*(x)$ and $\nabla h(x) = 0$.*

We also note that the computation of these directions does *not* require to compute the matrices $\nabla_{11}^2 G(z, x)$ and $\nabla_{21}^2 G(z, x)$: we only need to compute their product with a vector, which can be computed at a cost similar to that of computing a gradient.

The framework we propose is summarized in Algorithm 1. It consists in following a joint update rule in $(z, v, x)$ that follows directions $D_z^t, D_v^t$ and $D_x^t$ that are unbiased estimators of $D_z, D_v, D_x$. The first and most important remark is that whereas $\nabla h$ cannot be written as a sum over samples, the directions $D_z, D_v$ and $D_x$ involve only simple sums, since their expressions are "linear" in $F$ and $G$:

$$D_z(z, v, x) = \frac{1}{n} \sum_{i=1}^{n} \nabla_1 G_i(z, x) \ , \tag{7}$$

$$D_v(z, v, x) = \frac{1}{n} \sum_{i=1}^{n} \nabla_{11}^2 G_i(z, x) v + \frac{1}{m} \sum_{j=1}^{m} \nabla_1 F_j(z, x) \ , \tag{8}$$

$$D_x(z, v, x) = \frac{1}{n} \sum_{i=1}^{n} \nabla_{21}^2 G_i(z, x) v + \frac{1}{m} \sum_{j=1}^{m} \nabla_2 F_j(z, x) \ . \tag{9}$$

It is therefore straightforward to derive unbiased estimators of these directions. In [30], the authors considered one particular case of our framework, where each direction is estimated by using the STORM variance reduction technique (see [12]). Taking a step back by proposing the framework summarized in Algorithm 1 opens the way to potential new algorithms that implement other techniques that exist in stochastic single level optimization. In what follows, we study two of them.

## 2.1  First example: the SOBA algorithm

The simplest unbiased estimator is obtained by replacing each mean by one of its terms chosen uniformly at random, akin to what is done in classical single-level SGD. We call the resulting algorithm SOBA (StOchastic Bilevel Algorithm). To do so, we choose two independent random indices $i \in [n]$ and $j \in [m]$ uniformly and estimate each term coming from $G$ using $G_i$ and each term coming from $F$ using $F_j$. This gives the unbiased **SOBA directions**

$$D_z^t = \nabla_1 G_i(z^t, x^t) \ , \tag{10a}$$

$$D_v^t = \nabla_{11}^2 G_i(z^t, x^t) v^t + \nabla_1 F_j(z^t, x^t) \ , \tag{10b}$$

$$D_x^t = \nabla_{21}^2 G_i(z^t, x^t) v^t + \nabla_2 F_j(z^t, x^t) \ . \tag{10c}$$

This provides us with a first algorithm, SOBA, where we plug Equations (10a) to (10c) in Algorithm 1. We defer its analysis to the next section. Importantly, we use different step sizes for the update in $(z, v)$ and for the update in $x$. We use the same step size in $z$ and in $v$ since the inner problem and the linear system have similar conditioning, which is that of $\nabla_{11}^2 G(z^t, x^t)$. The need for a different step size for the outer and inner problem is clear: both problems can have a different conditioning.

An important remark for SOBA is that all the stochastic directions used are computed at the same point $z^t, v^t$ and $x^t$ with the same indices $(i, j)$. The update of $z$, $v$ and $x$ can thus be performed in parallel instead of sequentially, benefiting from hardware parallelism. Moreover, this enables to share the computations between the different directions. This is the case in hyperparameters selection where $G_i(z, x) = \ell_i(\langle z, d_i \rangle) + \frac{x}{2}\|z\|^2$, with $d_i$ a training sample, and $\ell_i$ that measures how good is the prediction $\langle z, d_i \rangle$. In this setting, we have $\nabla_1 G_i(z, x) = \ell_i'(\langle z, d_i \rangle) d_i + xz$ and $\nabla_{11}^2 G_i(z, x) v = \ell_i''(\langle z, d_i \rangle)\langle v, d_i \rangle d_i$. The prediction $\langle z, d_i \rangle$ can thus be computed only once to obtain both quantities. For more complicated models, where automatic differentiation is used to compute the different derivatives and Jacobian-vector products, we can store the computational graph only once to compute at the same time $\nabla_1 G_i(z, x), \nabla_{11}^2 G_i(z, x) v$ and $\nabla_{21}^2 G_i(z, x) v$, requiring only one backward pass, thanks to the $\mathcal{R}$ technique [37].

Finally, like all single loop bilevel algorithms, our method updates at the same time the inner and outer variable, avoiding unnecessary optimization of the inner problem when $x$ is far from the optimum.

## 2.2  Global variance reduction with the SABA algorithm

In classical optimization, SGD fails to reach optimal rates because of the variance of the gradient estimator. Variance reduction algorithms aim at reducing this variance, in order to follow directions that are closer to the true gradient, and to achieve superior practical and theoretical convergence.

In our framework, since the directions $D_z, D_v$ and $D_x$ are all written as sums of derivatives of $F_j$ and $G_i$, it is easy to adapt most classical variance reduction algorithms. We focus on the celebrated SAGA algorithm [13]. The extension we propose is called SABA (Stochastic Average Bilevel Algorithm). The general idea is to replace each sum in the directions $D$ by a sum over a memory, updating only one term at each iteration. To help the exposition, we denote $y = (z, x, v)$ the vector of joint variables. Since we have sums over $i$ and over $j$, we have two memories for each variable: $w_i^t$ for $i \in [n]$ and $\tilde{w}_j^t$ for $j \in [m]$, which keep track of the previous values of the variable $y$.

At each iteration $t$, we draw two random independent indices $i \in [n]$ and $j \in [m]$ uniformly and update the memories. To do so, we put $w_i^{t+1} = y^t$ and $w_{i'}^{t+1} = w_{i'}^t$ for $i' \neq i$, and $\tilde{w}_j^{t+1} = y^t$ and $\tilde{w}_{j'}^{t+1} = \tilde{w}_{j'}^t$ for $j' \neq j$. Each sum in the directions $D$ is then approximated using SAGA-like rules: given $n$ functions $\phi_{i'}$ for $i' \in [n]$, we define $S[\phi, w]_i^t = \phi_i(w_i^{t+1}) - \phi_i(w_i^t) + \frac{1}{n} \sum_{i'=1}^n \phi_{i'}(w_{i'}^t)$. This is an unbiased estimators of the average of the $\phi$'s since $\mathbb{E}_i\left[S[\phi, w]_i^t\right] = \frac{1}{n} \sum_{i=1}^n \phi_i(y^t)$.

With a slight abuse of notation, we call $\nabla_{11}^2 Gv$ the sequence of functions $(y \mapsto \nabla_{11}^2 G_i(z, x)v)_{i \in [n]}$ and $\nabla_{21}^2 Gv$ the sequence of functions $(y \mapsto \nabla_{21}^2 G_i(z, x)v)_{i \in [n]}$. We define the **SABA directions** as

$$D_z^t = S[\nabla_1 G, w]_i^t \ , \tag{11a}$$

$$D_v^t = S[\nabla_{11}^2 Gv, w]_i^t + S[\nabla_1 F, \tilde{w}]_j^t \ , \tag{11b}$$

$$D_x^t = S[\nabla_{21}^2 Gv, w]_i^t + S[\nabla_2 F, \tilde{w}]_j^t \ . \tag{11c}$$

These estimators are unbiased estimators of the directions $D_z$, $D_v$ and $D_x$. The SABA algorithm corresponds to Algorithm 1 where we use Equations (11a) to (11c) as update directions. When taking a step size $\gamma^t = 0$ in the outer problem, hereby stopping progress in $x$, we recover the iterations of the SAGA algorithm on the inner problem. In practice, the sum in $S$ is computed by doing a rolling average (see Appendix B for precision), and the quantities $\phi_i(w_i^t)$ are stored rather than recomputed: the cost of computing the SABA directions is the same as that of SGD. It requires an additional memory for the five quantities, of total size $n \times p + (n + m) \times (p + d)$ floats that can be reduced by using larger batch sizes. Indeed, if $b_{\text{in}}$ and $b_{\text{out}}$ are respectively the inner and the outer batch sizes, the memory load is reduced to $n_b \times p + (n_b + m_b) \times (p \times d)$ with $n_b = \lceil \frac{n}{b_{\text{inn}}} \rceil$ and $m_b = \lceil \frac{m}{b_{\text{out}}} \rceil$ which are smaller than the number of samples. This memory load can also be reduced in specific cases, for instance when $G$ and $F$ correspond to linear models, where the individual gradients and Hessian-vector products are proportional to the samples. In this case, we only store the proportionality ratio, reducing the memory load to $3n + 2m$ floats. Like for SOBA, the computations of the new quantities $\phi_i(w_i^{t+1})$ are done in parallel, thus benefiting from hardware acceleration and shared computations. Despite this memory load, using SAGA-like variance reduction instead of STORM as done in [30, 47, 28] has the advantage to bring the variance of the estimate directions to zero, enabling faster $O(\frac{1}{T})$ convergence.

In the next section, we show that SABA is fast. It essentially has the same properties as SAGA: despite being stochastic, it converges with fixed step sizes, and reaches the same rate of convergence as gradient descent on $h$.

## 3 Theoretical analysis

In this section, we provide convergence rates of SOBA and SABA under some classical assumptions. Note that, unlike most of the stochastic bilevel optimization papers, we work in finite sample setting rather than the more general expectation setting. Actually, SABA does not make any sense for functions that don't have a finite sum structure. However, we stress that SOBA could be studied in a more general setting to obtain the same bounds as here. Also, the finite sum setting is still interesting since doing empirical risk minimization is very common in practice in machine learning. The proofs and the constants in big-$O$ are deferred in Appendix C.

### 3.1 Background and assumptions

We start by stating some regularity assumptions on the functions $F$ and $G$.

**Assumption 3.1.** The function $F$ is twice differentiable. The derivatives $\nabla F$ and $\nabla^2 F$ are Lipschitz continuous in $(z, x)$ with respective Lipschitz constants $L_1^F$ and $L_2^F$.

Note that the above assumption is typically verified in the machine learning context, *e.g.,* when $F$ is the ordinary least squares (OLS) loss or the logistic loss.

**Assumption 3.2.** The function $G$ is three times continuously differentiable on $\mathbb{R}^p \times \mathbb{R}^d$. For any $x \in \mathbb{R}^d$, $G(\cdot, x)$ is $\mu_G$-strongly convex. The derivatives $\nabla G$, $\nabla^2 G$ and $\nabla^3 G$ are Lipschitz continuous in $(z, x)$ with respective Lipschitz constants $L_1^G$, $L_2^G$ and $L_3^G$.

Strong convexity and smoothness with respect to $z$ of $G$ are verified when $G$ is a regularized least-squares/logistic regression with a full rank design matrix, when the data is not separable for the

logistic regression. Moreover, the strong convexity ensures the existence and uniqueness of the inner optimization problem for any $x \in \mathbb{R}^d$.

**Assumption 3.3.** There exists $C_F > 0$ such that for any $x$ we have $\|\nabla_1 F(z^*(x), x)\| \leq C_F$.

This assumption, combined with the strong convexity of $G(\cdot, x)$, shows boundedness of $v^*$. This assumption holds, for instance, in the case of hyperparameters selection for a Ridge regression problem. Note that in Assumptions 3.1 and 3.2, we assume more regularity of $F$ and $G$ than in stochastic bilevel optimization literature (see for instance [19, 24, 26, 2]). It is necessary to get the smoothness of $v^*$ which will allow to adapt the proof of Chen et al. [9] and get tight convergence rates. The following lemma gives us some smoothness properties of the considered directions that will be useful to derive convergence rates of our methods.

**Lemma 3.4.** *Under the Assumptions 3.1 to 3.3, there exist constants $L_z$, $L_v$ and $L_x$ such that* $\|D_z(z, v, x)\|^2 \leq L_z^2 \|z - z^*(x)\|^2$, $\|D_v(z, v, x)\|^2 \leq L_v^2 (\|z - z^*(x)\|^2 + \|v - v^*(x)\|^2)$ *and* $\|D_x(z, v, x) - \nabla h(x)\|^2 \leq L_x^2 (\|z - z^*(x)\|^2 + \|v - v^*(x)\|^2)$.

In first order optimization, a fundamental assumption on the objective function is the smoothness assumption. In the case of vanilla gradient descent applied to a function $f$, it allows to get a convergence rate of $\|\nabla f(x^t)\|^2$ in $O(1/T)$, i.e. convergence to a stationary point [36]. The following lemma proved by Ghadimi and Wang [19, Lemma 2.2] ensures the smoothness of $h$.

**Lemma 3.5.** *Under the Assumptions 3.1 to 3.3, the function $h$ is $L^h$-smooth for some $L^h > 0$.*

The constant $L^h$ is specified in Appendix C.3. As usual with the analysis of stochastic methods, we define the expected norms of the directions $V_z^t = \mathbb{E}[\|D_z^t\|^2]$, $V_v^t = \mathbb{E}[\|D_v^t\|^2]$ and $V_x^t = \mathbb{E}[\|D_x^t\|^2]$, where the expectation is taken over the past. Thanks to variance-bias decomposition, they are the sum of the variance of the stochastic direction and the squared-norm of the unbiased direction. For SOBA, we use classical bounds on variances like those found for instance in [24]:

**Assumption 3.6.** There exists $B_z$, $B_v$ and $B_x$ such that for all $t$, $\mathbb{E}_t[\|D_z^t\|^2] \leq B_z^2(1 + \|D_z(z^t, v^t, x^t)\|^2)$ and $\mathbb{E}_t[\|D_v^t\|^2] \leq B_v^2(1 + \|D_v(z^t, v^t, x^t)\|^2)$ where $\mathbb{E}_t$ denotes the expectation conditionally to $(z^t, v^t, x^t)$.

For SOBA and SABA, we need to bound the expected norm of $D_x^t$. For SABA, this assumption allows to get a same sample complexity as SAGA for single level problems.

**Assumption 3.7.** There exists $B_x$ such that for all $t$, $\mathbb{E}_t[\|D_x^t\|^2] \leq B_x^2$.

Assumptions 3.6 and 3.7 are verified for instance, if all the $G_i$ and $\nabla_1 G_i$ have at most quadratic growth, and if $F$ has bounded gradients. They are also verified if the iterates remain in a compact set. Note that we do not assume that $G$ has bounded gradients, as this would contradict its strong-convexity. Finally, for the analysis of SABA, we need regularity on each $G_i$ and $F_j$:

**Assumption 3.8.** For all $i \in [n]$ and $j \in [m]$, the functions $\nabla G_i$, $\nabla F_j$, $\nabla_{11}^2 G_i$ and $\nabla_{21}^2 G_i$ are Lipschitz continuous in $(z, x)$.

### 3.2 Fundamental descent lemmas

Our analysis for SOBA and SABA is based on the control of both $\delta_z^t = \mathbb{E}[\|z^t - z^*(x^t)\|^2]$ and $\delta_v^t = \mathbb{E}[\|v^t - v^*(x^t)\|^2]$, Strong convexity of $G$ and smoothness of $z^*(x)$ and $v^*(x)$ allow to obtain the following lemma by adapting the proof of Chen et al. [9]. In what follows, we drop the dependency of the step sizes $\rho$ and $\gamma$ in $t$ for clarity.

**Lemma 3.9.** *Assume that* $\gamma^2 \leq \min\left(\frac{\mu_G L_*^2}{4B_x^2 L_{zx}^2}, \frac{\mu_G L_*^2}{8B_x^2 L_{vx}^2}\right)\rho$. *We have:*

$$\delta_z^{t+1} \leq \left(1 - \frac{\rho\mu_G}{4}\right)\delta_z^t + 2\rho^2 V_z^t + \beta_{zx}\gamma^2 V_x^t + \overline{\beta}_{zx}\frac{\gamma^2}{\rho}\mathbb{E}[\|D_x(z^t, v^t, x^t)\|^2]$$

$$\delta_v^{t+1} \leq \left(1 - \frac{\rho\mu_G}{8}\right)\delta_v^t + \beta_{vz}\rho\delta_z^t + 2\rho^2 V_v^t + \beta_{vx}\gamma^2 V_x^t + \overline{\beta}_{zx}\frac{\gamma^2}{\rho}\mathbb{E}[\|D_x(z^t, v^t, x^t)\|^2]$$

*where* $\beta_{zx} = \beta_{vx} = 3L_*^2$, $\overline{\beta}_{zx} = \frac{8L_*^2}{\mu_G}$, $\overline{\beta}_{vx} = \frac{16L_*^2}{\mu_G}$, $L_*$ *is the maximum between the Lipschitz constants of $z^*$ and $v^*$ (see Lemma C.1),* $\beta_{vz} = \frac{1}{\mu_G^3}(L_1^F \mu_G + L_2^G)^2$, $L_{zx}$ *and* $L_{vx}$ *are respectively the smoothness constants of $z^*$ and $v^*$.*

We insist that this result is obtained in general for Algorithm 1 with arbitrary unbiased directions. We can therefore invoke this lemma for the analysis of both SOBA and SABA. We use the smoothness of $h$ to get the following lemma, which is similar to [9, Lemma 1].

**Lemma 3.10.** *Let $h^t = \mathbb{E}[h(x^t)]$ and $g^t = \mathbb{E}[\|\nabla h(x^t)\|^2]$. We have*

$$h^{t+1} \leq h^t - \frac{\gamma}{2}g^t - \frac{\gamma}{2}\mathbb{E}[\|D_x(z^t, v^t, x^t)\|^2] + \frac{\gamma}{2}L_x^2(\delta_z^t + \delta_v^t) + \frac{L^h}{2}\gamma^2 V_x^t .$$

If $z^t = z^*(x^t)$, $v^t = v^*(x^t)$, that is $\delta_z$, $\delta_v$ both cancel and $D_x(z^t, v^t, x^t) = \nabla h(x^t)$, we get an inequality reminiscent of the smoothness inequality for SGD on $h$.

### 3.3 Analysis of SOBA

The analysis of SOBA is based on Lemmas 3.5 and 3.9. We have the following theorem, with fixed step sizes depending on the number of iterations:

**Theorem 1** (Convergence of SOBA, fixed step size)**.** Fix an iteration $T > 1$ and assume that Assumptions 3.1 to 3.7 hold. We consider fixed steps $\rho^t = \frac{\overline{\rho}}{\sqrt{T}}$ and $\gamma^t = \xi\rho^t$ with $\overline{\rho}$ and $\xi$ precised in the appendix. Let $(x^t)_{t\geq 1}$ the sequence of outer iterates for SOBA. Then,

$$\frac{1}{T}\sum_{t=1}^{T}\mathbb{E}[\|\nabla h(x^t)\|^2] = O(T^{-\frac{1}{2}}) .$$

As opposed to [24], we do not need that the ratio $\frac{\gamma}{\rho}$ goes to 0, which allows to get a complexity (that is, the number of call to oracles to have an $\epsilon$-stationary solution) in $O(\epsilon^{-2})$ better than the $\tilde{O}(\epsilon^{-\frac{5}{2}})$ they have. Also, note that this rate is the same as the one of SGD for non-convex and smooth objective [18, 8]. We obtain a similar rate using decreasing step sizes:

**Theorem 2** (Convergence of SOBA, decreasing step size)**.** Assume that Assumptions 3.1 to 3.7 hold. We consider steps $\rho^t = \overline{\rho}t^{-\frac{1}{2}}$ and $\gamma^t = \xi\rho$. Let $x^t$ the sequence of outer iterates for SOBA. Then,

$$\inf_{t\leq T}\mathbb{E}[\|\nabla h(x^t)\|^2] = O(\log(T)T^{-\frac{1}{2}}) .$$

As for SGD, SOBA suffers from the need of decreasing step sizes to get actual convergence because of the variance of the estimation on each directions. On the other hand, the analysis of SABA leverages the dynamic of all three variables, resulting in fast convergence with fixed step sizes.

### 3.4 SABA: a stochastic method with optimal rates

In what follows, we denote $N = n + m$ the total number of samples. The following theorem shows $O(N^{\frac{2}{3}}T^{-1})$ convergence for the SABA algorithm in the general case where we only assume smoothness of $h$. Our analysis of SABA is inspired by the analysis of single-level SAGA by Reddi et al. [40].

**Theorem 3** (Convergence of SABA, smooth case)**.** Assume that Assumptions 3.1 to 3.3 and 3.7 to 3.8 hold. We suppose $\rho = \rho'N^{-\frac{2}{3}}$ and $\gamma = \xi\rho$, where $\rho'$ and $\xi$ depend only on $F$ and $G$ and are specified in appendix. Let $x^t$ the iterates of SABA. Then,

$$\frac{1}{T}\sum_{t=1}^{T}\mathbb{E}[\|\nabla h(x^t)\|^2] = O\left(N^{\frac{2}{3}}T^{-1}\right) .$$

To prove the theorem, the idea is to control the distance from the memory to the current variables. We define $S^t = \frac{1}{n}\sum_{i=1}^{n}\|y^t - w_i^t\|^2 + \frac{1}{m}\sum_{j=1}^{m}\|y^t - \tilde{w}_j^t\|^2$ . In appendix, we show that we can find scalars $\phi_s, \phi_z, \phi_v > 0$ such that the quantity $\mathcal{L}^t = h^t + \phi_s S^t + \phi_z\delta_z^t + \phi_v\delta_v^t$ satisfies $\mathcal{L}^{t+1} \leq \mathcal{L}^t - \frac{\gamma}{2}g^t$. Summing these inequalities for $t = 1 \ldots T$ and using the fact that $\mathcal{L}^t$ is lower bounded demonstrates the theorem.

Note that the step sizes are constant with respect to the time, but they scale with $N^{-\frac{2}{3}}$. As a consequence, the sample complexity is $O(N^{\frac{2}{3}}\epsilon^{-1})$ which is analogous of the one of SAGA for

non-convex single level problems [40]. This is better than the sample complexity of Algorithm 1 with full batch directions, which is $O(N\epsilon^{-1})$. Hence, with SABA, we get the best of both worlds: the stochasticity makes the scaling in $N$ of the sample complexity goes from $N$ in full batch mode to $N^{\frac{2}{3}}$ for SABA, and the variance reduction makes the scaling in $\epsilon$ goes from $\epsilon^{-2}$ for SOBA to $\epsilon^{-1}$ for SABA. Our experiments in Section 4 confirm this gain.

Furthermore, if we assume that $h$ satisfies a Polyak-Łojasiewicz (PL) inequality, we recover linear convergence. Recall that $h$ has the PL property if there exists $\mu_h > 0$ such that for all $x \in \mathbb{R}^d$, $\frac{1}{2}\|\nabla h(x)\|^2 \geq \mu_h(h(x) - h^*)$ with $h^*$ the minimum of $h$.

**Theorem 4** (Convergence of SABA, PL case). Assume that $h$ satisfies the PL inequality and that Assumptions 3.1 to 3.3 and 3.7 to 3.8 hold. We suppose $\rho = \rho' N^{-\frac{2}{3}}$ and $\gamma = \xi \rho' N^{-1}$, where $\rho'$ and $\xi$ depend only on $F$ and $G$ and are specified in appendix. Let $x^t$ the iterates of SABA and $c' \triangleq \min\left(\mu_h, \frac{1}{16P'}\right)$ with $P'$ specified in the appendix. Then,

$$\mathbb{E}[h^T] - h^* = (1 - c'\gamma)^T (h^0 - h^* + C^0)$$

where $C^0$ is a constant specified in appendix that depends on the initialization of $z, v, x$ and memory.

The proof is similar to that of the previous theorem: we find coefficients $\phi_s, \phi_z, \phi_v$ such that $\mathcal{L}^t = h^t + \phi_s S^t + \phi_z \delta_z^t + \phi_v \delta_v^t$ satisfies the inequality $\mathcal{L}^{t+1} \leq (1 - c'\gamma)\mathcal{L}^t$, which is then unrolled. Note that in the case where we initialize $z$ and $v$ with $z^0 = z^*(x^0)$, $v^0 = v^*(x^0)$, and the memories $w_i^0 = w^0$, $\tilde{w}_j^0 = w^0$ for all $i, j$, the constant $C^0$ cancels and the bound simplifies to $\mathbb{E}[h(x^T)] - h^* \leq (1 - c'\gamma)^T(h(x^0) - h^*)$.

Just like classical variance reduction methods in single-level optimization, this theorem shows that our method achieves linear convergence under PL assumption on the value function. To the best of our knowledge, our method is the first stochastic bilevel optimization method that enjoys such property. We note that the PL hypothesis is more general than $\mu_h$-strong convexity of $h$ – it is a necessary condition for strong convexity.

We see here the importance of *global* variance reduction. Indeed, using variance reduction only on $z$ and SGD on $x$ would lead to sub-linear convergence in $x$. This would be the case even with a perfect estimation of $z^*(x)$. Similarly, using variance reduction only on $x$ and SGD on $z$ would lead to sub-linear convergence in $z$, and hence in $x$. Using global variance reduction with respect to each variable as we propose here is the only way to achieve linear convergence. We now turn to experiments, where we find that our method is also promising from a practical point of view.

## 4 Experiments

Here we compare the performances of SOBA and SABA with competitor methods on different tasks.

The different methods being compared are stocBiO [26], AmiGO [2], FSLA [30], MRBO [47], TTSA [24], BSA [19] and SUSTAIN [28]. A detailed account of the experiments is provided in Appendix B. [1]

### 4.1 Hyperparameters selection

The first task we perform is hyperparameters selection to choose regularization parameters on $\ell^2$ logistic regression. Let us denote $((d_i^{\text{train}}, y_i^{\text{train}}))_{1 \leq i \leq n}$ and $((d_i^{\text{val}}, y_i^{\text{val}}))_{1 \leq i \leq m}$ the training and the validation sets. In this case, the inner variable $\theta$ corresponds to the parameters of the model, and the outer variable $\lambda$ to the regularization. The functions $F$ and $G$ of the problem (1) are the logistic loss, with $\ell^2$ penalty for $G$, that is to say $F(\theta, \lambda) = \frac{1}{m}\sum_{i=1}^{m} \varphi(y_i^{\text{val}}\langle d_i^{\text{val}}, \theta\rangle)$ and $G(\theta, \lambda) = \frac{1}{n}\sum_{i=1}^{n} \varphi(y_i^{\text{train}}\langle d_i^{\text{train}}, \theta\rangle) + \frac{1}{2}\sum_{k=1}^{p} e^{\lambda_k}\theta_k^2$ where $\varphi(u) = \log(1 + e^{-u})$. We fit a binary classification model on the IJCNN1[2] dataset. Here, $n = 49\,990$, $m = 91\,701$ and $p = 22$.

The suboptimality gap is plotted in Figure 2a for each method. The lowest values are reached by SABA. Moreover, SABA is the only single-loop method that reaches a suboptimality below $10^{-3}$. SOBA reaches a quite high final value but slightly better than TTSA and FSLA. The gap between

---

[1]The code of the benchmark is available at https://github.com/benchopt/benchmark_bilevel and the results are displayed in https://benchopt.github.io/results/benchmark_bilevel.html.
[2]https://www.csie.ntu.edu.tw/~cjlin/libsvmtools/datasets/binary.html

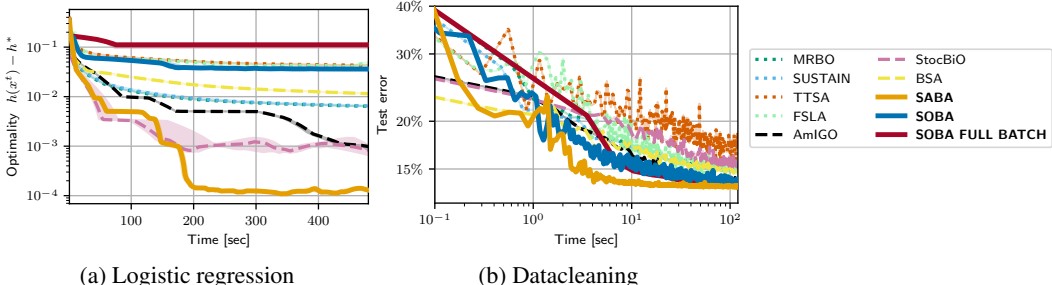

(a) Logistic regression      (b) Datacleaning

Figure 2: Comparison of SOBA and SABA with other stochastic bilevel optimization methods. For each algorithm, we plot the median performance over 10 runs. In both experiments, SABA achieves the best performance. The dashed lines are for one loop competitor methods, the dotted lines are for two loops methods and the solid lines are the proposed methods. **Left**: hyperparameter selection for $\ell^2$ penalized logistic regression on IJCNN1 dataset , **Right**: data hyper-cleaning on MNIST with $p = 0.5$ corruption rate.

SOBA and SABA highlights the benefits of variance reduction: it gives us a lower plateau and the fixed step sizes enable faster convergence.

## 4.2 Data hyper-cleaning

The second task we perform is data hyper-cleaning introduced in [16] on the MNIST[3] dataset. The data is patitioned into a training set $(d_i^{\mathrm{train}}, y_i^{\mathrm{train}})$, a validation set $(d_i^{\mathrm{val}}, y_i^{\mathrm{val}})$, and a test set. The training set contains 20000 samples, the validation set 5000 samples and the test set 10000 samples. The targets $y$ take values in $\{0, \ldots, 9\}$ and the samples $x$ are in dimension 784. Each sample in the training set is *corrupted* with probability $p$: a sample is corrupted when we replace its label $y_i$ by a random label in $\{0, \ldots, 9\}$. Samples in the validation and test sets are not corrupted. The goal of datacleaning is to train a multinomial logistic regression on the train set and learn a weight per training sample, that should go to 0 for corrupted samples. This is formalized by the bilevel optimization problem (1) with $F(\theta, \lambda) = \frac{1}{m} \sum_{i=1}^{m} \ell(\theta d_i^{\mathrm{val}}, y_i^{\mathrm{val}})$ and $G(\theta, \lambda) = \frac{1}{n} \sum_{i=1}^{n} \sigma(\lambda_i) \ell(\theta d_i^{\mathrm{train}}, y_i^{\mathrm{train}}) + C_r \|\theta\|^2$ where $\ell$ is the cross entropy loss and $\sigma$ is the sigmoid function. The inner variable $\theta$ is a matrix of size $10 \times 784$, and the outer variable $\lambda$ is a vector in dimension $n_{\mathrm{train}} = 20000$.

For the estimated parameters $\theta$ during optimization, we report in Figure 2b the test error, *i.e.*, the percent of wrong predictions on the testing data. We use for this experiment a corruption probability $p = 0.5$. In general, the error decreases quickly until it reaches a final value. We observe that our method SABA outperforms all the other methods by reaching faster its smallest error, which is smaller than the ones of the other methods. For SOBA, it reaches a lower final error than stocBiO and BSA. In appendix, we provide other convergence curves, and find that for higher values of $p$, SABA is still the fastest algorithm to reach its final accuracy. Overall, we find that among all methods, even those that implement variance reduction (that is FSLA, MRBO, SUSTAIN, SABA), SABA is the one that demonstrates the best empirical performance.

## 5 Conclusion

In this paper, we have presented a framework for bilevel optimization that enables the straightforward development of stochastic algorithms. The gist of our framework is that the directions in Equations (4) to (6) are all written as simple sums of samples derivatives. We leveraged this fact to propose SOBA, an extension of SGD to our framework, and SABA, an extension of SAGA to our framework, which both achieve similar convergence rates as their single level counterparts. Finally, we think that our framework opens a large panel of potential methods for stochastic bilevel optimization involving techniques of extrapolation, variance reduction, momentum and so on.

## Acknowledgments and Disclosure of Funding

We thank Othmane Sebbouh, Zaccharie Ramzi and Benoît Malézieux for their precious comments. The authors acknowledge the support of the ANER RAGA BFC. SV acknowledges the support of

---

[3]http://yann.lecun.com/exdb/mnist/

the ANR GraVa ANR-18-CE40-0005. This work is supported by a public grant overseen by the French National Research Agency (ANR) through the program UDOPIA, project funded by the ANR-20-THIA-0013-01 and DATAIA convergence institute (ANR-17-CONV-0003).

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
