# OpenReview forum: "A framework for bilevel optimization that enables  stochastic and global variance reduction algorithms"
_NeurIPS.cc/2022/Conference — NeurIPS 2022 Accept_

### Official Review · Reviewer_HH9R · 2022-06-23

**Rating:** 6
**Confidence:** 3
**Soundness:** 3 good
**Presentation:** 4 excellent
**Contribution:** 3 good

**Summary:**

In this paper, the author consider a finite-sum stochastic bilevel optimization problem:
$$\min_x F(x,z)\qquad s.t. \qquad z = \arg\min_{z'} G(x,z')$$
where $F(x,z) = \frac{1}{m} \sum_{i=1}^m F_i(x,z)$, $G(x,z) = \frac{1}{n} \sum_{j=1}^n G_j(x,z)$, and $G$ is strongly convex.
For $h(x) = F(x,z^*(x)), z^*(x):=\arg\min_{z'} G(x,z')$, it is known that the gradient of $h$ can be written as
$$\nabla h(x) = \nabla_x F(x,z^*(x)) - \nabla_{xz}G(z^*(x),x)\nabla_{zz}G(z^*(x),x)^{-1}\nabla_zF(x,z^*(x)).$$
Due to the finite sum feature of the objective function, it may not be easy to estimate the matrix inverse in the above formula. However, one should notice that $\nabla_{zz}G(z^*(x),x)^{-1}\nabla_zF(x,z^*(x)) = \arg\min_v \frac{1}{2}v^T\nabla_{zz}G(z^*(x),x)v - v^T\nabla_zF(x,z^*(x))$, the authors propose to update $v$ and $z$ by one step of the gradient descent while updating $x$ by one step of ``gradient descent'' where the inverse-vector product is replaced with $v$.

Compared to many previous algorithms, the proposed framework is matrix-inversion free and is very convenient to incorporate the SGD or SAGA variance reduction technique.

The reviewer think the proposed framework for stochastic bi-level optimization is convenient and  flexible. Most importantly, it is conceptually simple and implementation friendly.  However, the current work also has some room to improve. For example, for a typical finite-sum/stochastic optimization problem, the efficiency measure should be sample complexity or oracle complexity, instead of the iteration complexity that is presented in this work. Finally, the sample complexity dependence of SABA on $m+n$ is linear, which is a bit confusing to me since SAGA usually improve the dependence to $(m+n)^{2/3}$.

Overall, the paper does provide some novel and interesting ideas. However, the theoretical result does not justify the reason for using variance reduction. If the authors can answer this question, the reviewer is very willing to adjust to a higher rating.

**Questions:**

Major issue.
1. Please discuss the sample complexity in the paper, in particular the dependence on m and n. This is because when we use sample average approximation to construct the empirical objective function, $m,n$ are often very large (In fact they are $\Omega(\epsilon^{-1})$, where $\epsilon$ is for measuring $\|h(x)\|^2$)
2. The authors should carefully justify why SAGA scheme does not provide any theoretical improvement over full batch method.

Minor issue.
1. The term ``global'' in the title is confusing, I'm not sure what this global means.
2. In the appendix, in the table that list the complexity of the algorithms, a ``)'' is missing for SABA.


**Limitations:**

NA. This is a theoretical paper.

**Strengths And Weaknesses:**

Strength \# 1: The framework is new and is conceptually simple and is very flexible to incorporate different variance reduction techniques.
Strength \# 2: The whole algorithm is Hessian inverse free, which is easy to implement. (Though this is not the only work that is matrix inverse free)
Strength \# 3: The paper is very clear and well organized. Very well-written and easy to follow.


Weakness \# 1: The main theorems in the main paper only discuss iteration complexity instead of sample complexity.
Weakness \# 2: The sample complexity in the appendix for SABA  is $O((m+n)\epsilon^{-1})$, which seems exactly the same as the full batch deterministic version of the algorithm. Theoretically, this does not provide any advantage of using SAGA variance reduction scheme.
I'm confused why SAGA doesn't improve the factor $(m+n)$ to $(m+n)^{2/3}$.

---

> ### Author Response · Authors · 2022-08-02
> **Answer to reviewer HH9R**
>
> Thank you for the feedback. We appreciated that you pointed out the clarity, the flexibility and the simplicity of our framework.
>
> **Sample complexity of SABA:** Indeed, the dependance in $N = n + m$ in the sample complexity of SABA was in $N$ in the submitted version of the paper, which did not achieve any theoretical improvement from the full batch method. Nevertheless, as indicated in the general comment, we managed to improve the analysis of SABA **leading to $O(N^{\frac23}\epsilon^{-1})$ sample complexity for SABA**. This provides a theoretical improvement from the full batch method and explains the gap between SABA and SOBA FULL BATCH in the experiments.
>
> **Global variance reduction:** The term “global” refers to the fact that we perform the variance reduction globally for all the variables and not separately. As explained in the text, if we do variance reduction in only one variable and use SOBA-like updates for the others, we get slower convergence rate. So, the **global variance reduction allows us to get fast convergence rate**.

---

> > ### Comment · Reviewer_HH9R · 2022-08-08
> > **Response to the comments**
> >
> > I thank the authors for making the improvements. Since my only concern is resolved. I will raise my rating to 6.

---

> ### Comment · Area_Chair_CXLb · 2022-08-08
> **Acknowledge / engage with the rebuttal**
>
> Dear reviewer,
>
> Can you read the author's rebuttal, check if it addresses your concerns, and react to it?
>
> It is important to acknowledge this work by the authors and to respect it.
>
> Best,
> AC

---

### Official Review · Reviewer_wFQ9 · 2022-07-12

**Rating:** 7
**Confidence:** 3
**Soundness:** 3 good
**Presentation:** 3 good
**Contribution:** 2 fair

**Summary:**

This paper proposes a simple framework for solving bilevel optimization. The framework involves only 3 unbiased estimation in each iteration. The authors provide theoretical convergence guarantees for both SGD version and variance reduction version based on SAGA. Experimental results are provided showing their superior performance.

**Questions:**



**Limitations:**



**Strengths And Weaknesses:**

Strengths:
1. The proposed framework is very simple and clear. The idea of simplifying solving bilevel optimization into three unbiased estimation makes the theory more intuitive and straightforward.
2. Many theoretical guarantees are provided, which makes the statements well-supported. To my best knowledge, the convergence analysis shows optimal convergence rate under corresponding settings.

Weaknesses:
1. From what I can see, the key idea of this framework is the way of estimating the Hessian-vector product, i.e. 'v' (equation (5) in the text). However, as the authors mention in line 92, this idea is not novel.
2. In the discussion of convergence, the comparison is made in terms of iteration complexity. But the experiments part only shows comparison in terms of running time.
2. The experiments are only performed on one dataset for each application.

---

> ### Author Response · Authors · 2022-08-02
> **Answer to reviewer wFQ9**
>
> We thank the reviewer for they comments that, among others, commends the clarity and the simplicity of our method. The second weakness dealing with the choice of the comparisons made in the experiments is discussed in the general comment.
>
> **Introduction of $v$:** it is indeed not novel since it can be found in [1] and [2]. Nevertheless, the novelty of our paper lies more in **the generality of the framework** than in the introduction of $v$ itself. Note that [2] is a specific case of this framework using a STORM variance reduction technique only on the outer problem and [1] is out of our framework because it performs several steps in $z$ and $v$. Moreover, **we propose an adaptation of SAGA** which, to the best of our knowledge, has not been done in the literature of bilevel optimization and achieves fast convergence rates.
>
> **Experiments on only on dataset per application:** The datacleaning task with MNIST is classical in the stochastic bilevel optimization literature (see e.g. [2] or [3]). The hyperparameter selection for $\ell^2$-regularized logistic regression is also classical in the literature but it is usually done with the 20newsgroups dataset. We think that this dataset is not adapted for stochastic algorithms because the number of features is much higher than the number of samples (130,107 features and 18,846 samples). For that reason, we chose to perform the task on the IJCNN1 dataset for which stochastic algorithms are more suited (141,691 samples in total and 22 features). **We added in the Appendix B.5 an hyperparameter selection additional experiment on the Covtype dataset** which has 581, 012 samples, 7 classes and 54 features.
>
> [1] Michael Arbel and Julien Mairal. Amortized Implicit Differentiation for Stochastic Bilevel Optimization. In *International Conference on Learning Representations (ICLR)*, 2022.
>
> [2] Junyi Li, Bin Gu, and Heng Huang. A Fully Single Loop Algorithm for Bilevel Optimization without Hessian Inverse. In *Proceedings of the Thirty-sixth AAAI Conference on Artificial Intelligence*, AAAI’22, 2022.
>
> [3] Kaiyi Ji, Junjie Yang, and Yingbin Liang. Bilevel optimization: Convergence analysis and enhanced design. In *International Conference on Machine Learning (ICML)*, 2021

---

> > ### Comment · Reviewer_wFQ9 · 2022-08-09
> > **Response to authors**
> >
> > I thank the authors for the detailed response and improvements on the revision. My concerns are resolved and I increase my rating from 6 to 7.

---

> ### Comment · Area_Chair_CXLb · 2022-08-08
> **Acknowledge / engage with the review**
>
> Dear reviewer,
>
> Can you read the author's rebuttal, check if it addresses your concerns, and react to it?
>
> It is important to acknowledge this work by the authors and to respect it.
>
> Best,
> AC

---

### Official Review · Reviewer_AcuN · 2022-07-18

**Rating:** 8
**Confidence:** 4
**Soundness:** 3 good
**Presentation:** 4 excellent
**Contribution:** 3 good

**Summary:**

This work studies fully single-loop stochastic algorithms for bilevel optimization problems where the inner problem is strongly convex and the inner and outer objective are smooth and given by a finite sum. In particular, it provides a unified framework where outer ($x$) and inner ($z$) variables  together with the variable used to solve a linear system in the bilevel gradient expression ($v$), evolve jointly. The authors propose 2 methods: SOBA and SABA. SOBA  updates $x, z, v$ similarly to single-level SGD and uses two-timescales decreasing step-sizes for the inner and outer variable, while SOBA uses variance reduction with constant step-sizes, similarly to SAGA, thus taking advantage of the finite-sum structure of the problem. The authors prove $O(T^{2/5})$ and $O(1/T)$ stationary point rates, where $T$ is the number of iterations, for SOBA and SABA respectively. Furthermore, SABA converges linearly to the solution when on bilevel problems satisfying the PL-condition. Experiments in optimizing one regularization parameter per feature on IJCNN1 and data hyper-cleaning on MNIST show that SABA outperforms several other bilevel methods introduced recently.


**Questions:**

I have the following questions and comments.

- Authors say that the sum to compute the variance reduced gradients in SOBA is done in practice using a rolling average (Lines 177-178). Is this the case also in the presented experiments? If so it should be stated more explicitly. How is the rolling average computed exactly? I could not find details on this in the Appendix and I think they should be included. The authors should comment on the fact that this approximation is not covered by the theoretical analysis and maybe do an experiment comparing the exact and approximate average.

- The authors should add the finite sum assumptions in the abstract to make it clearer, since it is not standard in the related literature on bilevel rates.

- Can the analysis of SOBA be improved following [1]? The main difference between the two methods is that [1] uses a Neumann series approximation for the hypergradient, while SOBA updates the linear system variable online. What are the challenges in obtaining the rates in [1]? Explaining this could strengthen the paper.

- Line 294-299: on the “real” sample complexity of SABA and SOBA in full batch mode. The authors say that the step-sizes in SABA are proportional to the inverse of the number of examples, which makes the sample complexity of SABA and SOBA using all the examples (full batch mode) equal, while empirical results show that SABA performs better. Isn’t this because the analysis does not take into account the correlations between the single sample gradients? I believe the stronger this correlation, the stronger the gap between SABA and SOBA full batch.

- Plots for the experiments show how the performance varies over time. It could be interesting to see how the performance varies also in terms of the number of single sample gradients and jacobian vector products, which is not dependent on the hardware used.

- The authors present rates also under the PL-condition for the outer objective. Are there some interesting bilevel applications where this condition holds? If so, an experiment under this setting could improve the paper.

- Figure 1 shows SOBA and SABA performance in a toy problem. I could not find any detail on the specific problem. I suggest either to remove Figure 1 or to add some, even brief, details in the main body and/or in the Appendix.



Minor:
- Line 59 and 61: I think there should be an expectation in the inequalities for the rates.

- Lines 93-96: reference to Hong et al.. Do the authors mean that only one element of the Neumann series is used? This is incorrect since they use more than one in that paper. I also found that paragraph a bit too fast.


[1] Chen, Tianyi, Yuejiao Sun, and Wotao Yin. "Closing the gap: Tighter analysis of alternating stochastic gradient methods for bilevel problems." Advances in Neural Information Processing Systems 34 (2021): 25294-25307.


**Limitations:**

Limitations are only partially addressed. The non standard finite sum assumption is properly discussed in the theoretical analysis section but could be anticipated in the abstract. The discrepancy between the theoretical and practical updates of SABA is mentioned briefly and not properly addressed (See Question section).


**Strengths And Weaknesses:**

Strengths:
1. The presented framework is clean and allows to analyze fully single-loop algorithms. Previous analysis except for [1] usually did not consider updating the linear system variable online and with one descent step. Furthermore [1] directly consider variance reduction while here an analysis of a simpler algorithm (SOBA) is also provided.
2. This is the first work exploiting the finite sum assumptions for this kind of bilevel problems and reaching the same rates as in single-level optimization.
3. Experimental comparison is well done.  SABA is promising also in terms of practical performance.
4. Very well written.

Weaknesses:
1. Some possible discrepancy between the proposed algorithm and how it is actually implemented: the average for the variance reduction is a rolling average in practice, and this is not covered by the theory.
2. Rate for SOBA is not optimal and it is not clear why.
3. Experiments are quite small in scale and can be slightly improved. No experiment under the PL-condition. No plots showing how the performance varies w.r.t. the number of single-sample gradients and hessian-vector products.

Some of the points are expanded in the Questions section.

[1] Li, Junyi, Bin Gu, and Heng Huang. "A fully single loop algorithm for bilevel optimization without hessian inverse." Proceedings of the AAAI Conference on Artificial Intelligence. Vol. 36. No. 7. 2022.

**Post authors’ response.**

The authors have thoroughly addressed my concerns in their revision and response. In particular, concerning weakness 2, they now achieve optimal rates for SOBA under additional smoothness assumptions which are still realistic. Therefore I accordingly increased the score from 7 to 8.

---

> ### Author Response · Authors · 2022-08-02
> **Answer to reviewer AcuN**
>
> We thank the reviewer for the constructive feedback and for highlighting the clarity of the framework. The questions about the analysis of SOBA and the plots of the experiments are treated in the general comment. We address here the remaining points raised in the review:
>
> **Rolling average:** This indeed deserves clarifications. With the notation of the paper $S[\phi, w]^t_i = \phi(w_i^{t+1}) - \phi_i(w^t_i) + \frac1n \sum_{i’=1}^n \phi_{i’}(w_{i’}^t)$ and we denote by $A_t = \frac1n \sum_{i’=1}^n \phi_{i’}(w_{i’}^t)$. By “Rolling average”, we mean that in practice we do not compute $A_{t+1}$ from scratch only using the stored gradients, implementing the formula $A_{t+1}=\frac1n \sum_{i=1}^n \phi_i(w^t_i)$. Instead we do $A_{t+1} \leftarrow A_t + \frac1n (\phi_i(w^{t+1}_i) - \phi_i(w_i^t))$. This is not an approximation : both methods **compute the exact same quantity**, but the rolling average is more efficient because it has $O(1)$ computational complexity while the naive method has $O(n)$ computational complexity. Note that this is what is done in the classical SAGA for single level problems in practice. We have clarified this in the supplementary material (l.559).
>
> **Mentioning the finite sum in the abstract:** Indeed, it is worth making the abstract more precise on this point and we have added it.
>
> **Analysis of SOBA:** The analysis of SOBA of the submitted version is inspired by [1] and leads to $O(T^{-\frac25})$ convergence rate which is worse than $O(T^{-\frac12})$ in [2]. As explained in the general comment, we managed to improve our descent lemmas following the proof technique in [2], **leading to $O(T^{-\frac12})$ convergence rate for SOBA**. The challenge was not in the introduction of $v$ but in getting a $\gamma^2$ factor with the variance of $D_x^t$ instead of $\frac{\gamma^2}\rho$ because it requires that the ratio $\frac\gamma\rho$ goes to zero to get convergence. Also, we mentioned that we needed some more regularity on $F$ and $G$ to get the result. So, **in comparison with [2], we need more regularity to get the smoothness of $v^{*}$, but we avoid $O(\log(T))$ Neumann iterations per outer iteration**.
>
> **PL assumption:** This is a good point. However, this result is still interesting since it shows that **SABA has the same behavior than SAGA under PL assumption**. This is another instance of such behavior.  Also, some authors have shown convergence results under strong convexity assumption, which is stronger than PL assumption. See for instance Theorem 3.1 in [3], Theorem 1 in [1] and Corollary 2 in [4].
>
> **Figure 1:** The experimental details about this figure were indeed scarce. It is the selection of the regularization parameter on a Ridge regression problem with 10 features, 750 training samples and 250 validation samples. We have added these details in the Appendix.
>
> **Scale of the experiments:** About the scale of the experiments, we would like to stress that **we have performed a grid search** for the hyperparameters of the different methods, and **for each element of the grid, each method have been run 10 times**. For the experiment on hyperparameter selection on IJCNN1, we tried 63 combinations of hyperparameters for each method, and it took 1400 CPU hours. For the datacleaning task, we tried 121 combinations of hyperparameters and it took 2420 CPU hours. We also added in the appendix an experiment on the problem of hyperparameter selection for an $\ell^2$-regularized multiclass logistic regression with the covtype dataset.  We also tried 63 combinations of hyperparameter for each optimizer and the experiment took 525 CPU hours.
>
> [1] Mingyi Hong, Hoi-To Wai, Zhaoran Wang, and Zhuoran Yang. A Two-Timescale Framework for Bilevel Optimization: Complexity Analysis and Application to Actor-Critic. *preprint ArXiv 2007.05170*, 2021.
>
> [2] Tianyi Chen, Yuejiao Sun, and Wotao Yin. Closing the Gap: Tighter Analysis of Alternating Stochastic Gradient Methods for Bilevel Problems. In *Advances in Neural Information Processing Systems (NeurIPS)*, 2021.
>
> [3] Saeed Ghadimi and Mengdi Wang. Approximation Methods for Bilevel Programming. *preprint ArXiv 1802.02246*, 2018.
>
> [4] Michael Arbel and Julien Mairal. Amortized Implicit Differentiation for Stochastic Bilevel Optimization. In *International Conference on Learning Representations (ICLR)*, 2022.

---

> > ### Comment · Reviewer_AcuN · 2022-08-07
> > **Response to the authors**
> >
> > I thank the authors for the great effort put in the response and paper revision. I updated the review and increased the score accordingly.
> >
> > I have the following last wish, which will not negatively affect the score if not fulfilled. The new results are obtained by imposing additional regularity conditions which are somewhat unique to this work. Although I agree these are reasonable conditions, I would really appreciate if the authors would include, also in the appendix, a more detailed discussion on the results obtained without those conditions, i.e. the ones obtained in the original version of the paper. This would make it clearer to readers what exactly is the gain obtained with those conditions and further strengthen the paper.

---

> > > ### Author Response · Authors · 2022-08-08
> > > **Answer to reviewer AcuN**
> > >
> > > Thank you for updating your review and for your suggestion. We agree that it is worth mentioning what rate we can expect if we stick with the usual regularity assumptions on $F$ and $G$. We will add it in the Appendix of the final version.

---

### Author Response · Authors · 2022-08-02
**General comment to all reviewers and area chairs**

We first would like to thank all the reviewers for their remarks and for finding our framework *”clean”* (​​Rev. AcuN), *”clear”* (Rev. wFQ9, HH9R) and *”conceptually simple and is very flexible”* (Rev. HH9R).
We start by addressing two points that have been raised by several reviewers:

**Improvement of the analysis of SOBA and SABA:** Two reviewers asked if the analysis of SOBA and SABA can be improved. We are happy to share that we managed to do that, adapting our descent lemmas following the proof technique of [1]. **We get $O(T^{-\frac12})$ convergence rate for SOBA and $O(N^{\frac23}T^{-1})$ convergence rate for SABA ($T$ beeing the number of iterations and $N = n+m$ the total number of samples). This leads to a sample complexity that is the same as in the single-level case for SGD (respectively for SAGA).** Note that this improvement requires a more regularity of $F$ and $G$: we need to assume $F$ twice differentiable with Lipschitz Hessian (instead of only once differentiable with Lipschitz gradient) and $G$ three times differentiable with Lipschitz continuous third order derivative (instead of twice differentiable with Lipschitz Hessian). This additional regularity enables us to get the smoothness of $v^*$ and then, adapt the inequality we get for $\delta_z^t$ to $\delta_v^t$. Nevertheless, these new regularity assumptions still hold when $F$ is the Ordinary Least Squares loss or the logistic loss and when $G$ is a regularized least squares or logistic loss. We included the modified convergence rates, assumptions and proofs in the revised version.

**Choice of the plots for the experiments:** Indeed, our experiments show the performances over time because we wanted to compare the total complexity of the methods in practice. But we agree that sample complexity is important and that’s why we added plots of performance with respect to the number of calls to individual gradients or Hessian-vector products in the Appendix.


[1] Tianyi Chen, Yuejiao Sun, and Wotao Yin. Closing the Gap: Tighter Analysis of Alternating Stochastic Gradient Methods for Bilevel Problems. In *Advances in Neural Information Processing Systems (NeurIPS)*, 2021.

---

### Comment · Area_Chair_CXLb · 2022-08-03
**Discussion period**

Thanks to all reviewers and authors for their work on this submission.

As the discussion period starts, I want to make sure that reviewers have read the author's response, and if needed react to it.

This can be done either by communicating with authors or in private conversation within the reviewing team.

---

> ### Comment · Senior_Area_Chairs · 2022-08-06
> **Please address rebuttal comments**
>
> Dear Reviewers,
>
> Thanks for your work reviewing this paper. There are only a few days left for the discussion period.
>
> As the AC has already mentioned, you **must read the rebuttal**, and have any form of interaction with the authors, simply out of respect for the work they put addressing your comments.
>
> Hence we kindly ask you to read and comment **ASAP** on this new content.
>
> SAC.

---

### Meta-Review · Area_Chair_CXLb · 2022-08-23

**Recommendation:** Accept
**Confidence:** Certain

**Metareview:**

The main topic of this work is stochastic bilevel optimization. It provides an efficient algorithm for this task, and provides theoretical results in this setting.

The reviewers are unanimous that this is well-presented work of high quality and should be accepted, and so do I.

**Award:**

No

---

### Decision · Program_Chairs · 2022-09-14

Accept